# Aftereffect of Seven Years of Straw Handling on Soil Sustainability and Vitality

Arnoldas Jurys [1], Dalia Feizienė [1,*], Mykola Kochiieru [2], Renata Žvirdauskienė [3] and Virginijus Feiza [2]

1. Department of Plant Nutrition and Agroecology, Institute of Agriculture, Lithuanian Research Centre for Agriculture and Forestry, Instituto al. 1, Akademija, LT-58344 Kėdainiai, Lithuania; arnoldas.jurys@lammc.lt
2. Department of Soil and Crop Management, Institute of Agriculture, Lithuanian Research Centre for Agriculture and Forestry, Instituto al. 1, Akademija, LT-58344 Kėdainiai, Lithuania; mykola.kochiieru@lammc.lt (M.K.); virginijus.feiza@lammc.lt (V.F.)
3. Microbiology Laboratory, Institute of Agriculture, Lithuanian Research Centre for Agriculture and Forestry, Instituto al. 1, Akademija, LT-58344 Kėdainiai, Lithuania; renata.zvirdauskiene@lammc.lt
* Correspondence: dalia.feiziene@lammc.lt; Tel.: +370-688-940-14

**Abstract:** Straw, as organic material, contains macronutrients and a wide range of micronutrients. Properly treated straw can become a valuable source for soil improvement and crop nutrition needs. The field experiment was carried out at the Institute of Agriculture, Lithuanian Research Centre for Agriculture and Forestry, in 2014–2021 on Cambisol. On a shallow ploughless tillage background, eight treatments were investigated: chopped straw + ammonium nitrate (CSA), chopped straw + ammonium nitrate + NPK (CSA+F), chopped straw + microorganisms (CSM), chopped straw + microorganisms + NPK (CSM+F), straw removed, not fertilized (SR), straw removed, fertilized (SR+F), chopped straw, not fertilized (CS), chopped straw, fertilized (CS+F). We hypothesized that treatment of straw with microbiological products in combination with mineral NPK fertilizers is a more efficient technology than treatment/non-treatment of straw with ammonium nitrate, either with or without NPK fertilizers. The aim of this work was to investigate the aftereffects of seven years use of mineral NPK fertilizers and bioproducts containing soil bacteria and microscopic fungi (*Bacillus megaterium*, *Acinetobacter calcoaceticus*, and *Trichoderma reesei*) in combination with straw management on soil sustainability (soil C sources, soil water release characteristics, pore-size distribution, aggregate stability, crop yielding capability) and soil vitality ($CO_2$ exchange rate-NCER). It was revealed that NCER was highest in the treatment CSM+F (*Bacillus megaterium*, *Acinetobacter calcoaceticus*, and *Trichoderma reesei* + NPK). It was 32.95% higher than in CSA (chopped straw without fertilizers) and 45.34% higher than in CSA+F (chopped straw + ammonium nitrate + NPK). Bioproducts applied favored soil vitality in general by exhibiting higher soil microbiological activity. As a result, a healthy and more viable Cambisol produced a higher winter wheat grain yield.

**Keywords:** plant residue treatment; $CO_2$ emission; plant-growth-promoting microorganisms; microbial biodiversity

## 1. Introduction

Fertilization, crop rotation, or land-use change affect the chemical and biological properties of the soil over time [1]. In crop cultivation technologies, the plant's nutrient requirements are met through forms such as mineral fertilizers, animal manure, or both. The use of mineral fertilizers in agriculture has made a significant positive contribution to increasing crop yields and is one of the most popular agricultural management practices [2]. In the long run, excessive fertilization may cause severe agronomic and environmental problems, such as crop lodging, nutrient leaching, and groundwater quality deterioration on arable land [2]. By focusing on crop fertilizer application under intensive farming, the effects of soil microorganisms receive little attention when solving crop fertilization

problems [1]. The long-term sustainability of agricultural systems depends on the microorganisms in the soil, which are involved in the essential processes of soil formation and the cycling of nutrients [1–3]. The effectiveness of agricultural fertilizers depends on the microorganisms in the soil. It is important to note that the biodiversity and activity of soil microbial biomass respond to soil management practices such as the use of organic and mineral fertilizers, crop rotation, tillage and fallow, and land-use change. Organic fertilizers (e.g., straw) are known to contain many specific nutrients as well as high levels of organic matter and microelements. The incorporation of straw into the soil after harvesting is an integral part of crop-growing technology. Properly treated straw can become a valuable source of plant nutrients for crop nutrition needs. This process is essential to improving SOC sequestration and the physical and chemical properties of the soil [3]. The incorporation of a straw causes rapid stimulation of the soil microflora, resulting in the activity of the soil enzyme and the supply of organic matter, increasing soil organic carbon and nitrogen resources, restoring soil fertility, changing soil aggregate size distribution, and reducing greenhouse gas emissions [3,4].

## 2. Materials and Methods

### 2.1. Soil and Site Description

The field experiment was conducted at the Institute of Agriculture, Lithuanian Research Centre for Agriculture and Forestry (55°23′50″ N and 23°51′40″ E). The soil is Endocalcari-Epihypogleyic Cambisol (CMg-n-w-can) with a loam texture (52–54% sand (2.0–0.05 mm), 29–32% silt (0.05–0.002 mm), and 14–19% clay (<0.002 mm).

### 2.2. Experimental Design and Methodology

Shallow ploughless tillage was applied during seven experimental years (2014–2021). Integral crop protection measures were implemented in accordance with the requirements of the crops grown and meteorological conditions.

The rates of mineral fertilizers were calculated according to soil properties and the target grain yield. No manure has been applied. The intensive cereal-based crop rotation was implemented: winter wheat (*Triticum aestivum* L., target yield was 8 t/ha, $N_{215}P_{22}K_{118}$), spring barley (*Hordeum vulgare L. nutans*, target yield was 6 t/ha, $N_{149}P_5K_{92}$), field bean (*Vicia faba* L., target yield was 5 t/ha, $N_0P_{66}K_{92}$), spring wheat (*Triticum aestivum* L., target yield was 5 t/ha, $N_{102}P_0K_{40}$), spring triticale (target yield was 5 t/ha, $N_{137}P_{98}K_{60}$), ley consisting of a mixture of 7 grasses (*Pisum sativum* L.—35 %, *Lupinus angustifolius* L.—24 %, *Avena sativa* L.—22 %, *Vicia sativa* L.—13.6 %, *Raphanus sativus* L.—4 %, *Phacelia tanacetifolia* L.—1.4 %), and winter wheat (target yield was 8 t/ha, $N_{179}P_{141}K_{120}$). During the 2014–2021 period, there were incorporated total amounts of nutrients: N—782 kg/ha, $P_2O_5$—332 kg/ha, and $K_2O$—522 kg/ha. Nitrogen fertilizers were applied three times during the grain crop vegetation period.

A field experiment according to 2-factorial design (Table 1) was set up in four replications. In CDM treatment we use three plant-growth-promoting microorganisms (Table 2). The gross plot size and net harvested area of each individual plot were 30.0 and 22.0 $m^2$, respectively.

### 2.3. Sampling and Soil Analysis

Soil samples for physical and chemical analysis were collected after crop harvesting. Disturbed soil samples were collected from a depth of 0–10 cm using an auger of 2 cm in diameter in each individual plot. Undisturbed core samples were collected using stainless steel rings (diameter 5 cm, volume 100 $cm^3$) from the 5–10 cm soil depth in 4 replicates per treatment to determine dry bulk density (BD), pore-size distribution (PSD), water release characteristics (WRC), and plant available water content (PAW). Samples from the 0–5 cm depth were omitted from the investigations because of complicated sampling due to soil crumbling at the very topsoil layer. Total porosity (TP) was calculated from the soil particle density and BD of the samples. The soil particle density was determined using the pycnometer method. The sandbox method (using a sandbox with a negative water potential

of 0 to −100 hPa; a Sand/kaolin box with a negative water potential of −100 to −490 hPa; and a membrane apparatus box with a negative water potential of −982 to −15,500 hPa) was implemented [5]. Pore-size classes were estimated using soil-water retention curves. These water retention curves were determined by exposing the completely saturated samples to constant suction levels of −4, −10, −30, −50, −100, −300 and −15,500 hPa. In this paper, the basic classes of soil pores were taken into consideration, including aeration pores/macropores (>30 μm diameter), capillary pores/mesopores (from 0.2 to 30 μm diameter), and micropores (<0.2 μm diameter). The water content levels at −100 hPa and at −15,500 hPa were considered the field capacity (FC) and the permanent wilting point (WP), respectively. The amount of water between these two suctions was regarded as plant-available water (PAW) content. Soil-water aggregate stability was determined by the wet sewing method with an Eijkelkamp Agriseerch Equipment apparatus (Zevenaar, The Netherlands). Soil samples were taken at randomly selected sampling points at two places in each individual plot from the 0–10 and 10–20 cm layers. Net carbon exchange rate (NCER). A closed-chamber method was used to quantify soil surface $CO_2$ effluxes in crop stands. At study sites and within all treatments, measurements of soil surface $CO_2$ effluxes were obtained at the main development stages of crops. All measurements were taken between 11 a.m. and 3 p.m. to reduce the variability in $CO_2$ flux due to diurnal changes in temperature [6]. A portable infrared $CO_2$ analyzer (IRGA) attached to a data logger was implemented (LiCor–6400/XT, LI-COR Biosciences GmbH, Hamburg, Germany). A closed, dark chamber was mounted on top of the collar, which was inserted into the soil to a depth of 10.0 cm. The area of the chamber is 81 cm$^2$. The chamber hood was placed on the stationary collar at the soil surface for 2 min in each plot until $CO_2$ efflux reached a steady reading. The data were recorded in the data logger. The chamber covered areas between crop rows. The $CO_2$ efflux measurements were done in four replications for each treatment. Soil surface volumetric water content (Wsoil), soil temperature (T), and soil pore water electrical conductivity (ECp) were recorded at the 0–10 cm soil depth by a portable soil sensor (type WET-2 with an HH2 Moisture Meter; Delta-T Devices Ltd., Burwell, UK) near the LiCor–6400 chamber at the same time as $CO_2$ efflux measurements were made. The biology system and the Biolog–EcoPlate procedure were used to determine 31 carbon sources and the metabolic functional diversity of soil microorganisms [7].

**Table 1.** Design of the experiment.

| No. | Treatments |
|---|---|
| Factor A—Straw Handling | |
| CSA | Chopped straw incorporation + ammonium nitrate for decomposition * |
| CSM | Chopped straw incorporation + microorganisms for decomposition |
| SR | Straw removed |
| CS | Chopped straw incorporation |
| Factor B—fertilization | |
| 1 | Not fertilized |
| 2 (F) | Fertilization according to soil properties and target yield |

Note: * ammonium nitrate (10 kg N/t) was applied to chopped straw after cereal harvesting.

**Table 2.** Use of amendments in CSM treatment.

| Use of Amendments | |
|---|---|
| After harvesting—for decomposition of organic residues | During crop vegetation |
| *Trichoderma reesei* | *Acinetobacter calcoaceticus* + *Bacillus megaterium* |

Note: Doses of substances: *Trichoderma reesei*—100 mL/ha, *Bacillus megaterium*—100 mL/ha, *Acinetobacter calcoaceticus*—100 mL/ha.

## 3. Results

### 3.1. Soil Net CO$_2$ Exchange Rate

The soil net CO$_2$ exchange rate (NCER) was measured five times during vegetation (Figure 1). NCER was highest in the treatment CSM+F (*Bacillus megaterium*, *Acinetobacter calcoaceticus*, and *Trichoderma reesei* + NPK). It was 32.95% higher than in CSA (chopped straw without fertilizers) and 45.34% higher than in CSA+F (chopped straw + ammonium nitrate + NPK). The average values of the two treatments (CSA and CSA+F) were 23.58% lower than the average values of the highest ones (CSM and CSM+F). The treatments CSM, SR+F, and CS+F did not differ significantly from each other.

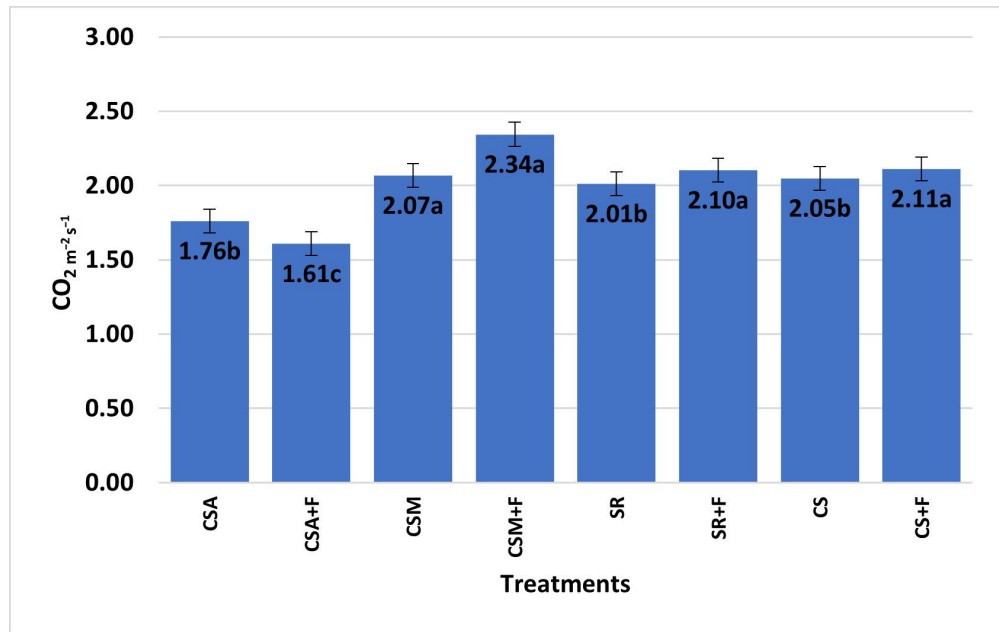

**Figure 1.** Mean soil net CO$_2$ exchange rate during winter wheat growing season in 2021. Note: data followed by the same letters are not significantly different at $p < 0.05$.

We can conclude that the higher soil net CO$_2$ exchange rate depended more on the microbiological amendments applied than on straw management itself. The treatments with ammonium nitrate revealed the lowest impact as compared to the other treatments investigated. The possible reason for this could be that the use of ammonium nitrate caused an acidifying effect on soil microorganisms [8]. Many microorganisms struggle to function properly under conditions of increased acidity. We assume that the reduced content of microorganisms caused a decrease in NCER.

### 3.2. Soil Temperature

Soil temperature was determined at the same time as NCER. It was also measured five times during vegetation. Soil temperature in experimental treatments averaged from 18.38 to 25.14 °C (Figure 2). The mean value of soil temperature in treatments CSA and CSA+F was 18.63 °C, and it was 16.14% lower than in other treatments.

In treatments where straw was removed (SR and SR+F), the mean soil temperature was the highest and reached 25.14 and 24.28 °C, respectively. The mean value of soil temperature in these treatments was higher by 16.14% compared to the other treatments investigated. It can be concluded that the chopped straw caused a significant reduction in soil temperature at the soil surface.

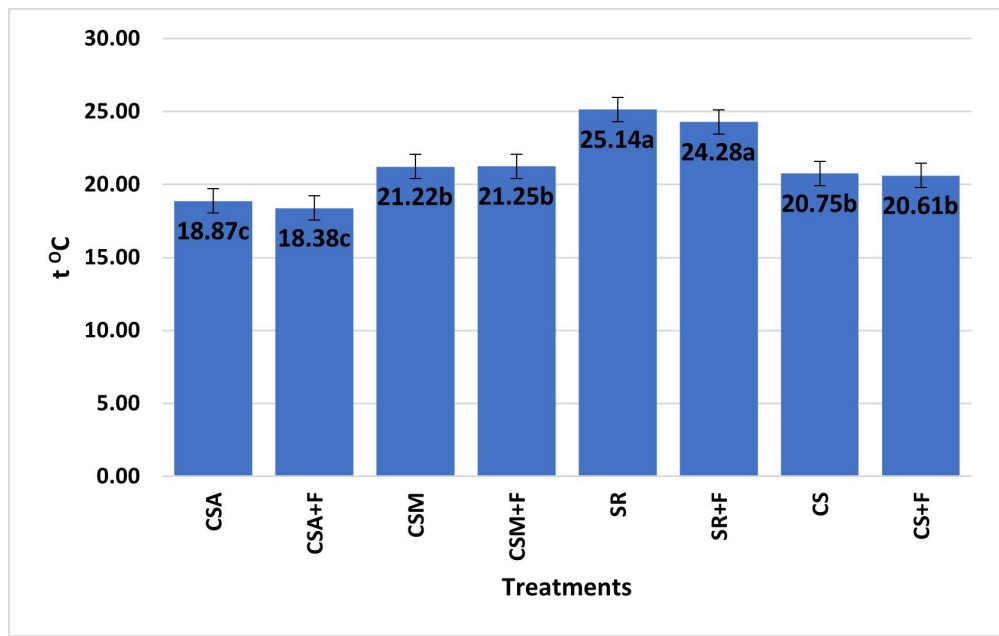

**Figure 2.** Mean soil temperature during winter wheat growing season in 2021. Note: data followed by the same letters are not significantly different at $p < 0.05$.

### 3.3. Soil Volumetric Water Content

Soil moisture is an important factor in the mineralization and humification of organic carbon, a transformation of nutrients into forms accessible to plants. In the 2021 growing season, the mean volumetric soil water content (VWC) varied from 10.4% to 13.5% (Figure 3). The lowest VWC was registered in treatment CSA—10.4% (chopped straw + ammonium nitrate)—and the highest in treatment CSM+F (*Bacillus megaterium*, *Acinetobacter calcoaceticus*, and *Trichoderma reesei* + NPK)—13.54%. In addition, soil VWC was very similar in all the treatments with the use of bioproducts, and these results were equivalent to the VWC with the use of ammonium nitrate. From the data obtained, we can summarize that the treatments investigated did not influence VWC.

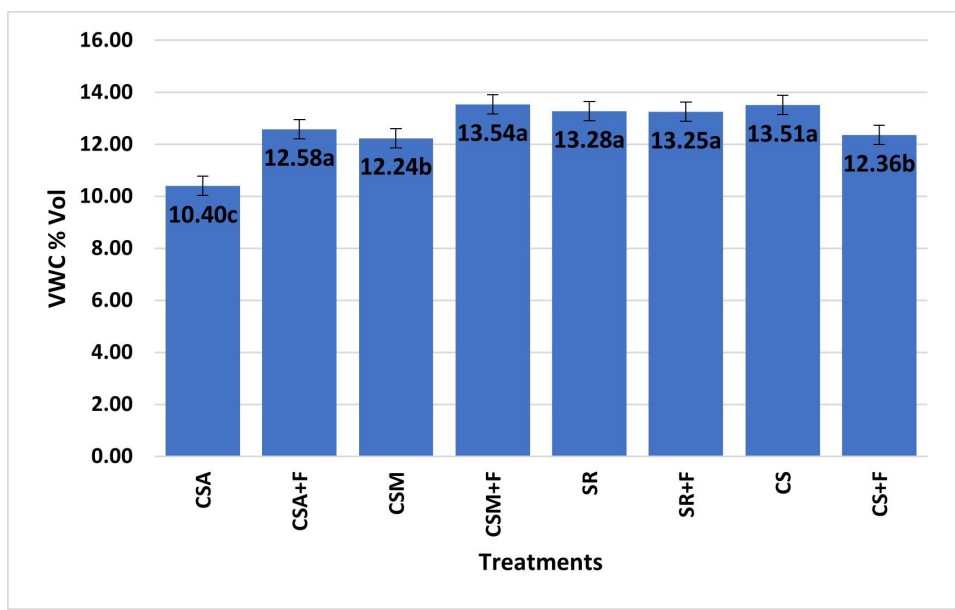

**Figure 3.** Mean soil volumetric water content during winter wheat growing season in 2021. Note: data followed by the same letters are not significantly different at $p < 0.05$.

### 3.4. Mean Soil Pore Water Electrical Conductivity

A higher electrical conductivity (ECp) of soil pores indicates a higher nutrient concentration and better nutrient availability to plants in the soil. The ECp varied considerably across all treatments (Figure 4). ECp was highly affected by mineral fertilizer application, while it was independent of the straw management system. In fertilized treatments, ECp varied from 87.54 to 95.35 mScm$^{-1}$, but differences were statistically unreliable. In unfertilized treatments, ECp was lower by 36.37% than in fertilized treatments and varied from 54.38 to 67.16 mScm$^{-1}$, but differences were also statistically unreliable. We can summarize that NPK fertilizers had a decisive effect on ECp. Moreover, the effectiveness of biological products sprayed to promote straw mineralization (CSM and CSM+F) was comparable to that of ammonium nitrate (CSA and CSA+F) for the ECp index.

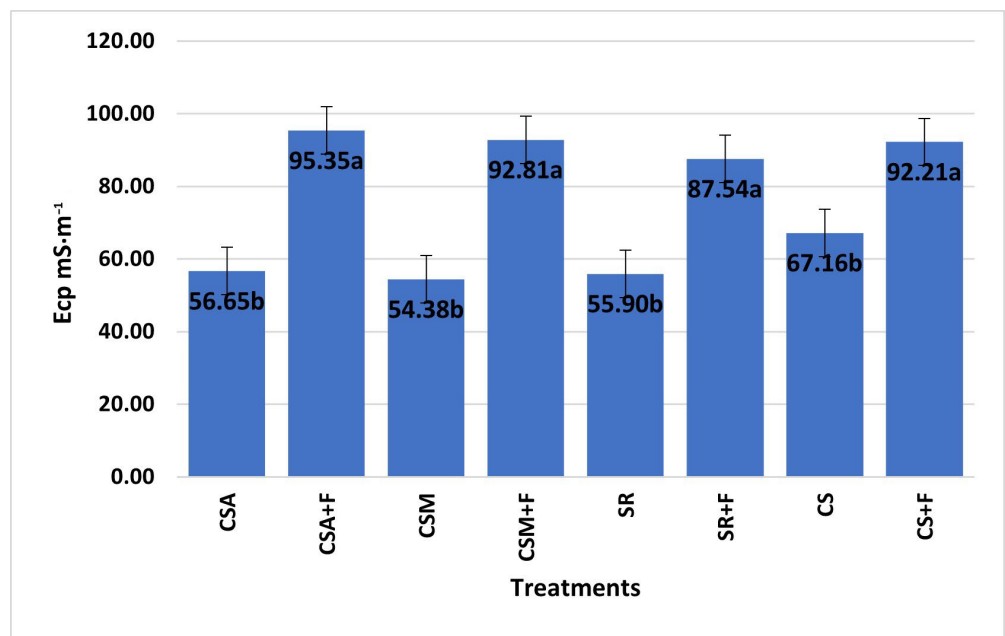

**Figure 4.** Mean soil pore water electrical conductivity during winter wheat growing season in 2021. Note: data followed by the same letters are not significantly different at *p* < 0.05.

### 3.5. The Influence of Management Practices on Soil Water Release Characteristics (WRC)

Soil water release characteristics (WRC) provide information on the available water range in soil for plants to grow in a sustainable way under dry weather conditions.

Straw handling methods caused dissimilar soil WRC within the 0–10 cm layer. Soil water content within the 0–10 cm layer revealed that the effect of residue handling on WRC at −4 hPa, −10 hPa, and −30 hPa was similar. Meanwhile, WRC at −50 hPa, −100 hPa, and −300 hPa was significantly higher in SR and SR+F treatments (straw removed from the field). Water retention at −15,500 hPa suction was similar in all treatments. Accordingly, the plant available water content (PAW) within the 0–10 cm layer of SR and SR+F treatments became 15% greater than in all treatments with residue returning (Figure 5). Such results can be explained by the response of pore-size distribution (PSD) to residue (straw) handling.

Despite the fact that straw increased the total porosity, the mesopore volume was reliably reduced compared to the treatment with straw removal (Figure 6). The volume of mesopores in SR and SR+F treatments (straw removed from the field) was 13% higher than in treatments where straw was returned. The distribution of macro- and microporosity was insignificant.

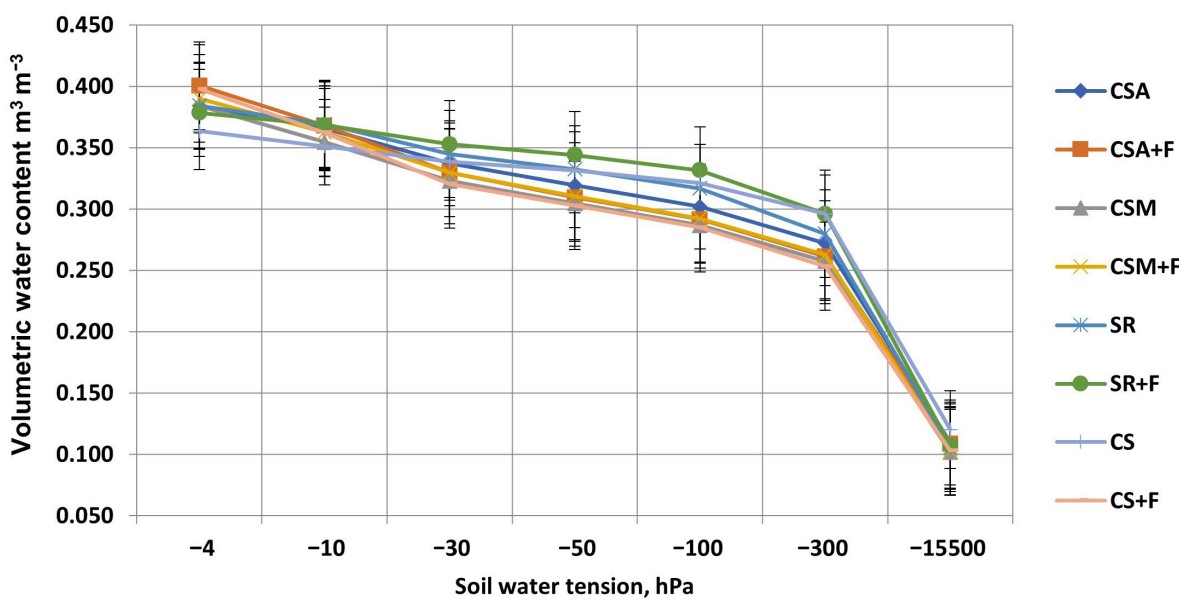

**Figure 5.** The impact of applied technology on soil water retention in the 5–10 cm layer.

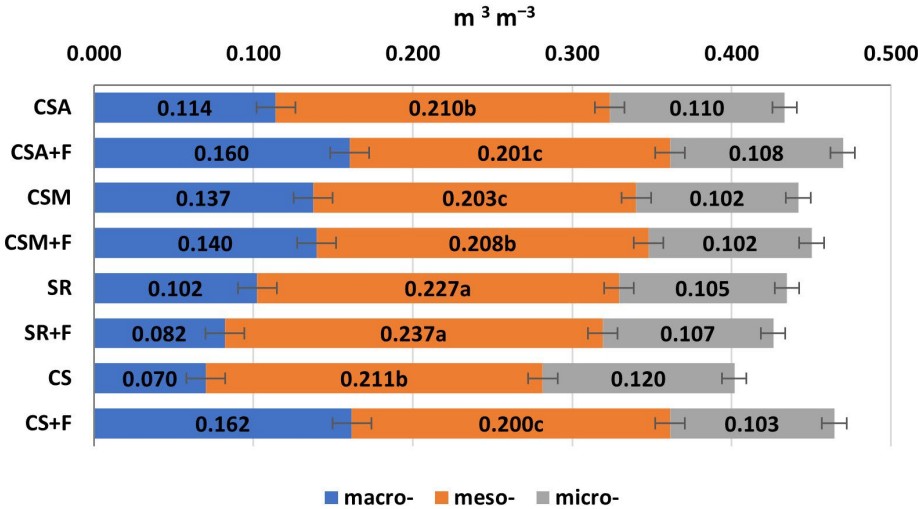

**Figure 6.** Soil pore size distribution in the 0–10 cm layer. Note: data followed by the same letters are not significantly different at $p < 0.05$.

Straw handling methods also caused dissimilar soil WRC within the 10–20 cm layer, but they were not as pronounced as at the 0–10 cm depth. Soil water content within the 0–10 cm layer revealed that the effect of residue handling on WRC at −4 hPa, −10 hPa, and −30 hPa was similar. It is important to notice that WRC at −50 hPa, −100 hPa, and −300 hPa was significantly higher in SR and SR+F treatments. Water retention at −15,500 hPa suction was similar in all the treatments. In addition, the plant available water content (PAW) within the 10–20 cm layer of SR and SR+F treatments was 7% greater than in all treatments with residue returning (Figure 7). The response of pore-size distribution (PSD) was a valuable tool to explain the effect of straw handling on soil pore geometry changes. It was revealed that the soil mesopore volume reliably decreased compared to the treatments where straw was removed (Figure 8). The volume of mesopores in SR and SR+F treatments (straw removed from the field) was 6% higher than in treatments where straw was returned. The distribution of macro- and microporosity was insignificant. We can assume that straw acted as a pore-clogging material within the overall arable layer, significantly decreasing mesoporosity [9].

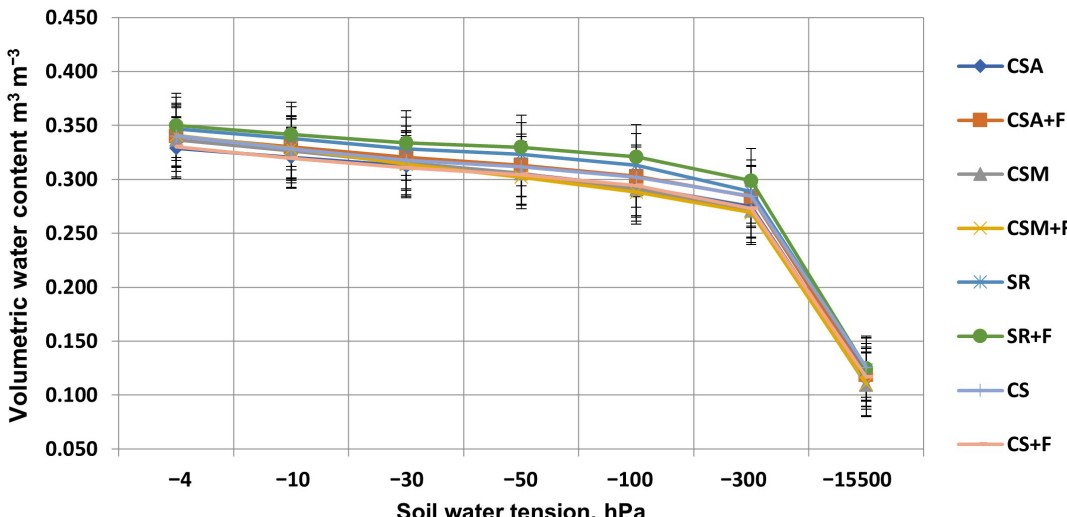

**Figure 7.** The impact of applied technology on soil water retention in the 15–20 cm layer. Note: data followed by the same letters are not significantly different at $p < 0.05$.

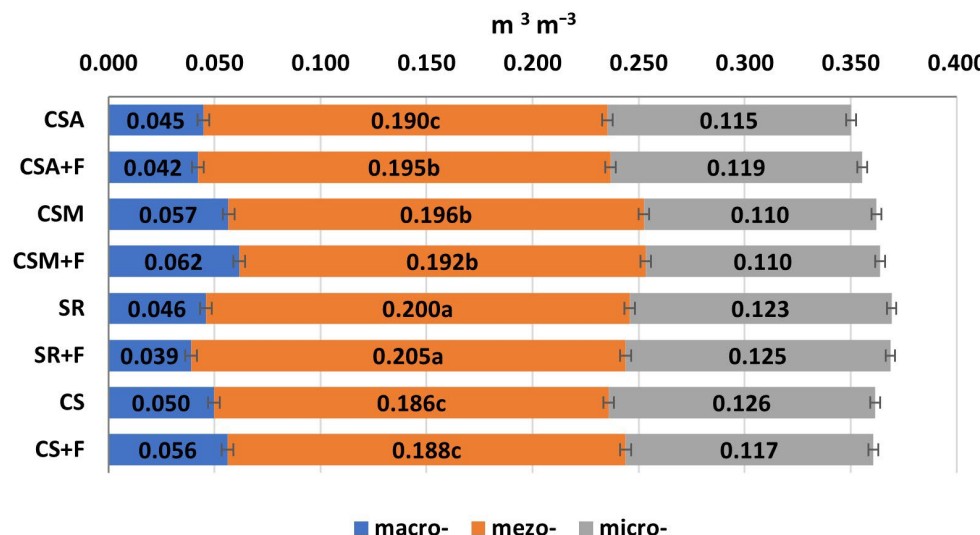

**Figure 8.** Soil pore size distribution in the 10–20 cm layer. Note: data followed by the same letters are not significantly different at $p < 0.05$.

### 3.6. Stability of Soil Aggregates

Soil aggregates comprise primary particles and binding agents and are the basic units of soil structure. The dynamics of all soil physical properties depend on the structure of the soil. Soil aggregates and the pores between them affect water movement and storage, aeration, erosion, biological activity, and productivity. Soil fertility is closely related to its structure. The formation and stabilization of soil aggregates are affected by the quality and amount of straw residue. Residues play very important roles in the process of aggregate formation. From an agronomic point of view, the most valuable aggregates are those that do not decompose when exposed to water and can retain their stability for a long time.

A good soil environment consists of durable crumbs with a diameter of 0.25–10 mm, but the most valuable are 0.25–5 mm.

The average stability of soil aggregates, in diameter 0.25–1 mm, within a 0–10 cm depth was 43.94%. In the treatments CSA+F, CSM, CSM+F, and CS, the stability of these soil aggregates was significantly higher than in other treatments. The mean stability of soil aggregates with a diameter >1 mm was 7.66%. The mean total stability of aggregates

(0–10 cm) in treatments reached 51.6%. In the treatments where straw was returned, the soil aggregate stability was higher than in the treatments where straw was removed (Figure 9).

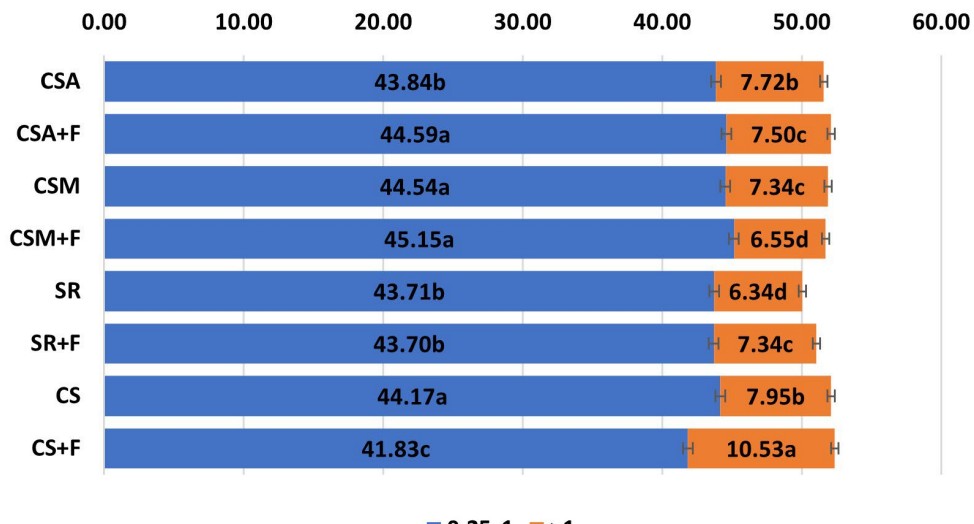

**Figure 9.** Stability of soil aggregates (%) in the 0–10 cm layer. Note: data followed by the same letters are not significantly different at $p < 0.05$.

The mean stability of soil aggregates with a diameter of 0.25–1 mm within a depth of 10–20 cm was 42.10%. In treatments CSA+F, CSM, and CSM+F, the stability of soil aggregates was significantly higher than in other treatments. The mean stability of aggregates larger than 1 mm within a 10–20 cm depth reached 8.47%. In treatment CSA, CSA+F, and CS+F, this index was significantly higher than in other treatments. The total stability of all aggregates (10–20 cm) reached 50.47% (Figure 10).

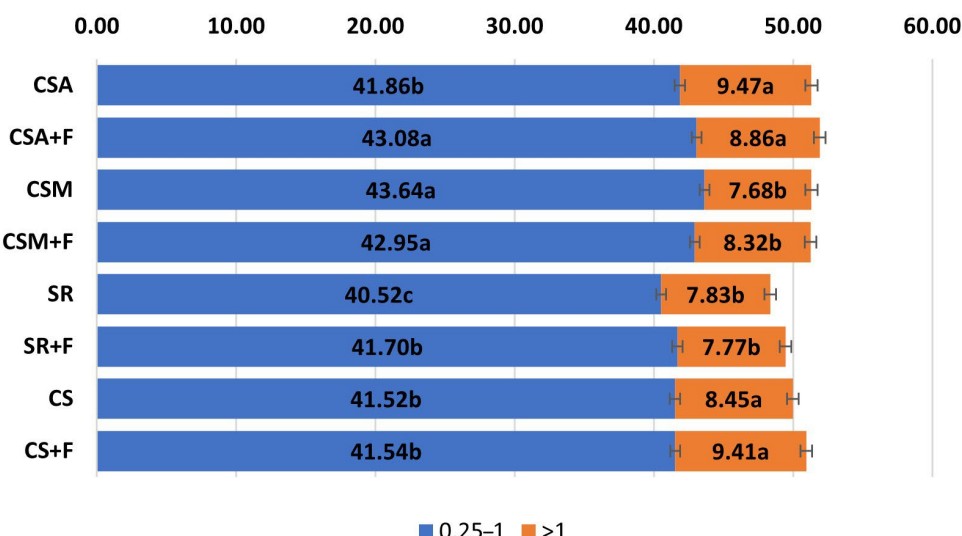

**Figure 10.** Stability of soil aggregates (%) in the 10–20 cm layer. Note: data followed by the same letters are not significantly different at $p < 0.05$.

We can assume that in both soil depths of the Ap horizon (0–10 and 10–20 cm soil layers), shallow incorporation of chopped straw, NPK fertilizers, bioproduct microorganisms, or ammonium nitrate application had a similar and positive effect on soil aggregate stability.

### 3.7. Relative Distribution of C Sources

In a soil environment, microorganisms are most likely to react to chemical and physical changes. The analysis of microorganism communities determines the pattern of reactions. It is known that microorganisms are one of the most important parts of nutrient metabolism. The EcoPlate analysis includes 31 carbon sources, which are used for community analysis. These carbon sources are divided into six main groups: carbohydrates, carboxylic acids, amino acids, polymers, amines, and miscellaneous.

Carbohydrates—polyfunctional organic compounds—are essential for living organisms. In organisms, carbohydrates are involved in biological processes, acting as energy sources or as parts of more complex organic compounds.

Carbohydrates were one of the dominant groups and averaged 24.8% of all carbon sources (Figure 11) in the experiment. A higher than mean carbohydrate value was established in the CSA+F, CSM, CSM+F, and SR+F treatments. The highest carbohydrate content was determined in the treatment CSA+F. It reached 28.9%. The lowest carbohydrate content was determined in treatment SR—22.2%.

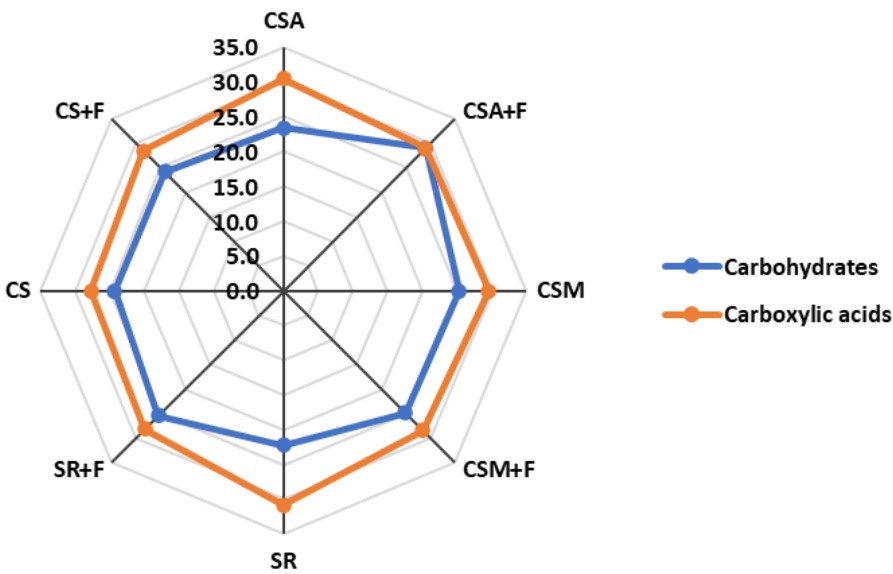

**Figure 11.** Level of consumption of carbohydrates and carboxylic acids during winter wheat growing season in 2021.

Changes in carboxylic acids depend mainly on seasonality, vegetation type, soil type, and depth. Carboxylic acids were the most dominant group of carbon sources and accounted for 29.1% of all carbon sources (Figure 11) in the experiment. Only in the treatments CSA, CSM, and SR-F has a higher content of carboxylic acids been identified than the experimental average. The largest carboxylic acid content was determined in the treatments SR and CSA. It was 30.8% and 30.4%, respectively. A somewhat lower content was determined in the treatment CSM—29.6%. The lowest carboxylic acid content was determined in treatment CS—27.7%.

Polymeric carbon sources increase particle adhesion. The incorporation of plant residues increases monosaccharide and polysaccharide polymers, microbiological metabolites, or simple carbohydrates. Microbial polysaccharides are one of the most important components in increasing the stability of aggregates, but the effectiveness of these polymers varies depending on microbiological abundance and diversity and the prevailing environmental conditions.

Polymeric carbon sources averaged from 11.3 to 14.6% of all carbon sources (Figure 12) in the experiment. In treatments SR, SR+F, CS, and CS+F, the content of polymers was higher than average. The largest amount of polymeric carbon sources was determined in

the treatment CS, which was 14.6%. The lowest amount of polymeric carbon sources was determined in the treatment CSA+F, which was 11.3%.

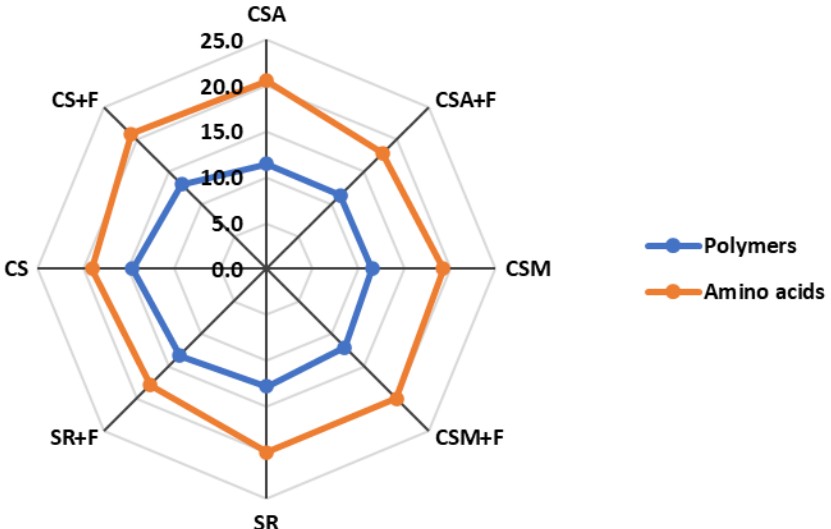

**Figure 12.** Level of consumption of polymers and amino acids during winter wheat growing season in 2021.

Amino acids are an important source of nitrogen for plants. Millions of proteins are known to be involved in vital processes. Amino acids reduce plant stress and improve photosynthesis. Amino acids averaged 19.4% of all carbon sources (Figure 12) in the experiment. In the treatments CSA, CSM+F, SR, and CS+F, there was a significantly higher content of amino acids compared to the average value. The largest content of amino acids was determined in the treatments CS+F and CSA. It was 20.9% and 20.5%, respectively. The lowest content of amino acids was determined in the treatments CSA+F and SR+F. It was 17.9%, respectively.

Amine augmentation in the soil is very important to mitigate $CO_2$ emissions. Amines are a major part of the factors controlling vital processes and a key element in amino acids and protein membranes. Amines and miscellaneous were the smallest groups of carbon sources and averaged 5.7% and 8.4% of all carbon sources, respectively (Figure 13) throughout the experiment. In the treatments CSM, CSM+F, SR, SR+F, and CS, the content of amines was significantly higher compared to the experiment average. The largest content of amines was determined in the treatment SR+F, and it was 6.5%. In the treatments of CSM and SR, it was 6.2%. The lowest content of amines was determined in the treatment CSA+F (4.5%).

In the treatments CSA, CSA+F, CSM+F, SR+F, and CS, more miscellaneous carbon sources were registered than the experimental average. The largest amount of miscellaneous carbon sources was determined in the treatments CSA, CSM+F, and SR+F (it was 8.7%). The lowest amount of miscellaneous carbon sources was in treatment SR (8.0%).

### 3.8. Substrate Consumption Potential, Richness, Shannon–Weaver Index

The substrate consumption potential (AWCD), Shannon–Weaver index (H), and richness index (R) were determined by the Biolog Eco Plates method. The AWCD and H indexes reflect the potential for use of carbon sources by soil microbial communities and are therefore important indicators of microbial activity. AWCD depended on the duration of the incubation period and the substances used (Figure 14). Changes in AWCD increased with an increasing incubation duration. The average OD value after 24 h was 0.29. In the treatments CSA, CSA+F, CSM, and SR+F, the OD values were somewhat higher than the average. The average OD value after 48 h was 1.14. In the treatments CSA, CSA+F, CSM, and SR+F, the OD values were also higher than the experimental average. The average

OD value after 72 h reached 1.55. In the treatments CSA+F, CSM, SR+F, and CS+F, the OD values were higher than the experimental average. The average OD value after 96 h was 1.76. In the treatments CSA+F, CSM, SR, and CS+F, the OD values were higher than the average.

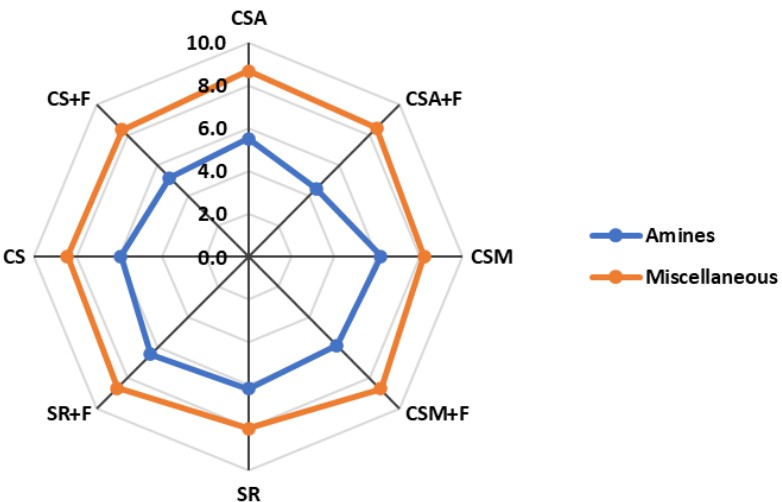

**Figure 13.** Level of consumption of amines and miscellaneous during winter wheat growing season in 2021.

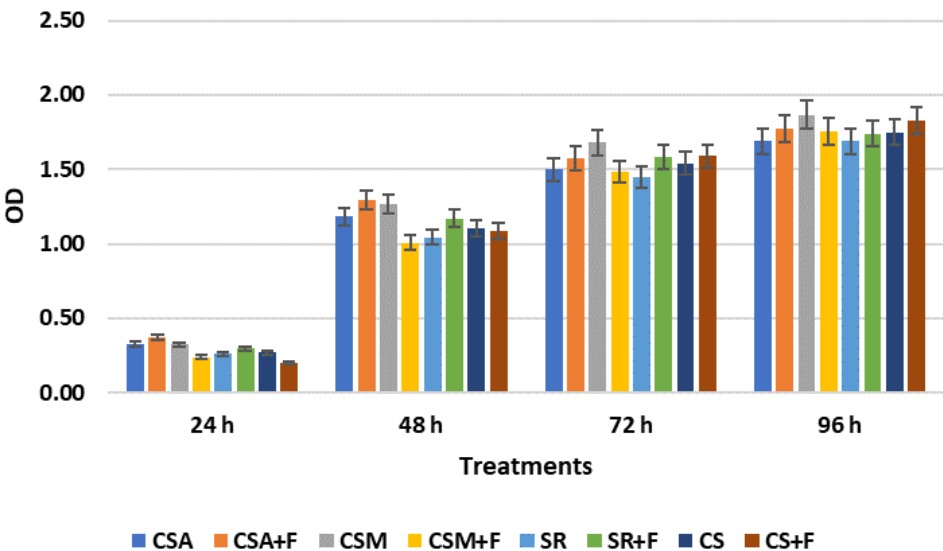

**Figure 14.** The influence of applied technologies on substrate consumption potential during winter wheat growing season in 2021.

The highest OD values were detected after 96 h of incubation in treatments CSM and CS+F (1.87 and 1.83, respectively). In the treatments CSA and SR-F, the OD values were the lowest (1.69) and differed from the maximum values by 8.23–10.50%.

The abundance index R depended on the duration of the incubation (Figure 15). The average OD value in the experiment after 24 h was 13.63. In the CSA, CSA+F, CSM, SR+F, and CS treatments, the OD values were higher than the average. The average OD value after 48 h was 22.33. In CSA+F, CSM, SR+F, and CS+F treatments, the OD values were also higher than the average.

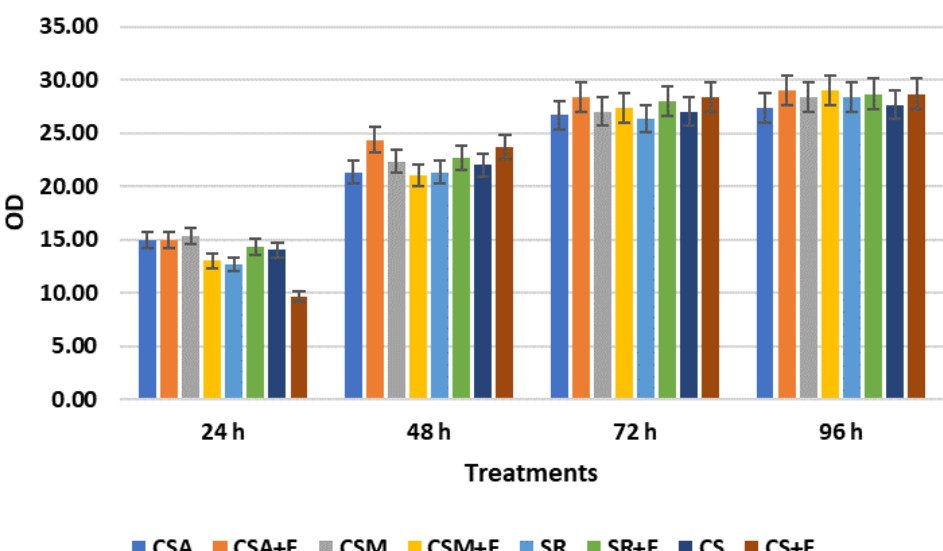

**Figure 15.** Influence of applied technology on richness during winter wheat growing season in 2021.

The average OD value after 72 h was 27.38. In CSA+F, SR+F, and CS+F treatments, the OD values were higher than the average. The average OD value after 96 h increased and reached 28.38. In the CSA+F, CSM+F, SR+F, and CS+F treatments, the OD values were higher than the average. The highest OD values were detected after 96 h of incubation in the treatments CSA+F and CSM+F (29.0). The lowest OD values were detected in the CSA and CS treatments—they were 27.33 and 27.67, respectively.

Changes in the Shannon-Weaver biodiversity index during the entire incubation period were marginal (Figure 16). The average OD value after 24 h was 2.90. In the CSA, CSA+F, CSM, and CSM+F treatments, the OD values were higher than the average. The average OD value after 48 h was 3.07. In the CSA+F, CSM, SR+F, and CS+F treatments, the OD values were higher than the average as well. The average OD value after 72 h was 3.22. In the CSA+F, CSM, SR+F, and CS+F treatments, the OD values were higher than the experimental average. The average OD value after 96 h amounted to 3.28. In the CSA+F, CSM, CSM+F, SR+F, and CS+F treatments, the OD values were higher than the average of this experiment.

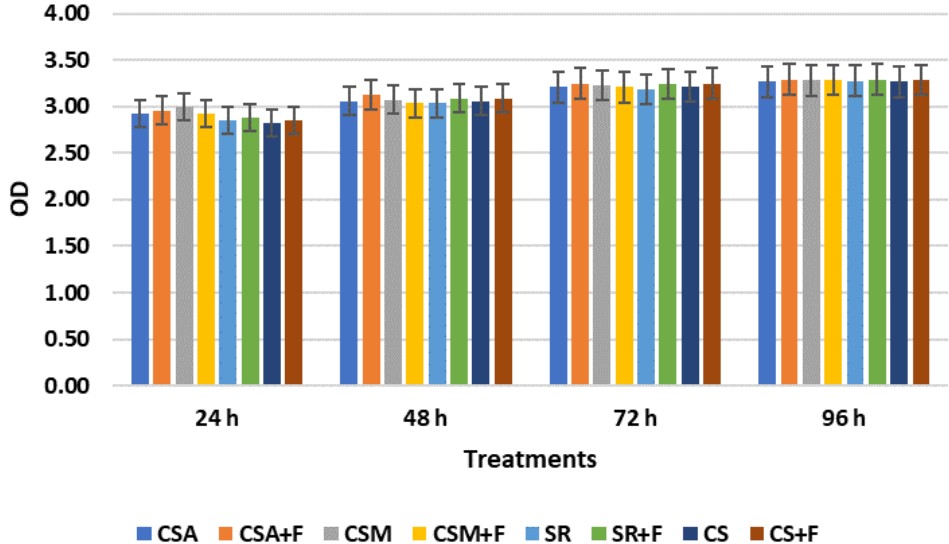

**Figure 16.** Influence of applied technology on Shannon-Weaver index during winter wheat growing season in 2021.

*3.9. The Influence of Different Plant Residue Management Technologies on Winter Wheat Grain Yield*

Data revealed the expected effect of a seven-year-long application of microbiological measures on plant productivity. Winter wheat grain yield in the experiment varied from 3.81 to 8.22 t/ha (Figure 17). Grain yield was mainly dependent on the application of mineral fertilizers. Fertilization increased yield by 69–116% compared to unfertilized crops. Traditional technology (CSA+F; chopped straw incorporation + ammonium nitrate for decomposition + NPK fertilization) caused a grain yield of 7.81 t/ha, which was 76% higher than without fertilization (CSA). The effectiveness of microbiological products (CSM+F) was significantly higher, as they caused a 116% higher grain yield compared to unfertilized crops (CSM). The effectiveness of fertilization in the treatment with straw removal (SR+F) was not so high; NPK fertilizers increased grain yield by 69% compared to unfertilized crops.

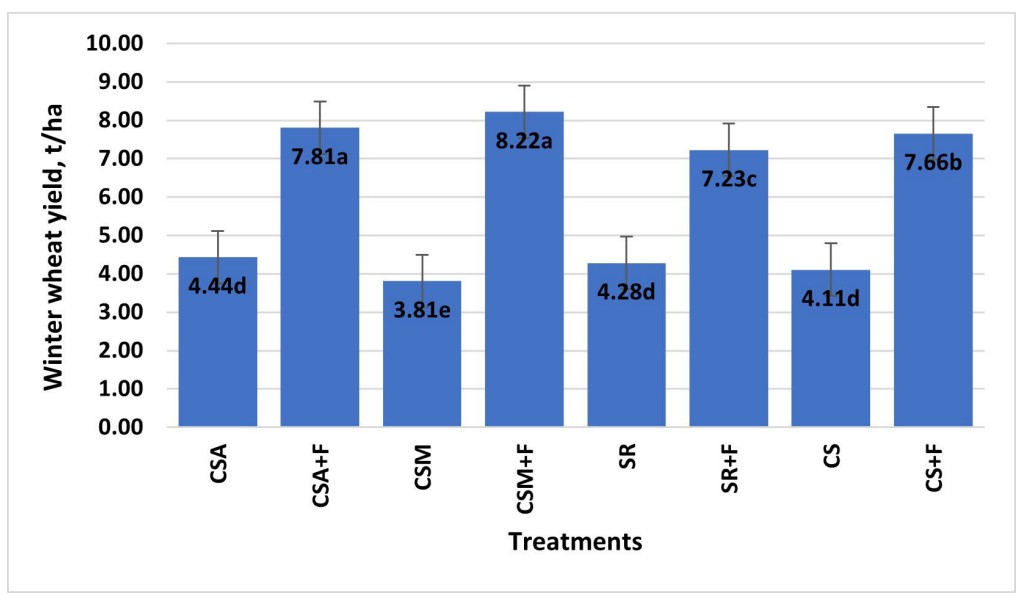

**Figure 17.** The impact of applied technology on crop grain yield. Note: data followed by the same letters are not significantly different at *p* < 0.05.

We can assume that NPK fertilizers had a significant effect on winter wheat yield, while the application of microbiological products additionally produced 0.41 t/ha of grain.

## 4. Discussion

Cereal straw, as an agricultural byproduct, can be recognized as a source for soil quality maintenance and crop yield improvement. Recently, many technologies have been developed for straw handling, of which the biological method has attracted growing interest because of its eco-friendly nature. However, different straw management technologies are very important and largely determine the profitability and environmental sustainability of farms [10].

In Lithuania, several straw management technologies are most widely used: (1) straw is removed from the field; (2) straw is chopped and incorporated into the soil; (3) the chopped straw is sprayed with N fertilizers to stimulate mineralization and then incorporated into the soil; and (4) a new and promising technology is the spraying of the chopped straw with a microbiological product to stimulate mineralization prior to incorporation. Straw burning is not an option, as it damages soil viability and environmental sustainability.

Biofertilizer is a substance containing live microorganisms that exhibit beneficial properties for soil quality, plant growth, and development. Two PGPR strains of Lithuanian origin and 1 PGPF strain were used in this study [11]. *Acinetobacter calcoaceticus* is a Gram-negative, rod-shaped (0.9–1.6 μm by 1.5–2.5 μm) in the intensive growth phase,

aerobic soil bacterium belonging to the genus *Acinetobacter*. *Bacillus megaterium* is a Gram-positive, rod-shaped (2.0–5.0 μm by 1.2–1.5 μm), mainly aerobic soil bacterium belonging to the genus *Bacillus*. It is one of the largest known bacteria. Both bacteria are known for their phosphorus-releasing and nitrogen-fixing properties [12–14]. *Trichoderma reesei* is a filamentous fungus widely distributed in places of soil and wood decomposition. This microscopic fungus is one of the main microorganisms used for the industrial production of biomass-degrading enzymes [15,16]. The saprophytic fungus of the genus *Trichoderma* obtains its nutrients from dead and decaying plant matter. *Trichoderma reesei* is known as a plant cell wall degrader that efficiently produces enzymes that degrade plant cell wall compounds such as cellulose, hemicellulose, and pectin [17].

Liu and others argued that the use of nitrogen fertilizers alone, in the long run, has significantly reduced the amount of microbial biomass, biodiversity, and enzyme activity compared to complex fertilization [1]. Therefore, the aim of this study was to investigate the combined effectiveness of straw management methods and mineral fertilizers. Our study has shown that the use of nitrogen fertilizers alone as a product to activate straw mineralization has, in the long run, significantly reduced substrate consumption potential, richness index, and biodiversity. However, the use of complex fertilization increased the richness index and biodiversity in all treatments, except the treatment using microbiological products. Meanwhile, the use of complex fertilization had different effects on C sources in straw management technologies. Such fertilization in CSA+F treatment significantly increases the content of carbohydrates; in SCM+F treatment, the content of polymers, amino acids, and miscellaneous; in SR+F treatment, the content of carbohydrates, polymers, amines, and miscellaneous; and in CS+F treatment, carboxylic acids and amino acids.

Chopped straw caused a reduction of soil temperature on the soil surface, while the investigative tools used and straw removal from the soil did not have a regular influence on VWC within the 0–10 cm soil layer. Even though chopped straw, NPK fertilizers, and microbiological products increased the aggregate's stability and total porosity, straw acted as a pore-clogging agent in the whole plough layer, significantly reducing mesoporosity and PAW content [9].

NPK fertilizers have a decisive effect on ECp. Moreover, the effectiveness of biological products sprayed to promote straw mineralization (CSM and CSM+F) was comparable to that of ammonium nitrate (CSA and CSA+F) for the ECp index. Wei and others find that the incorporation of wheat straw, in comparison with chemical fertilizers alone, had a very significant effect on the available nutrients in the 0–40 cm soil layer, especially in the top layer of soil (0–20 cm) [2]. We also found a trend: in treatments with straw returned (CSA, CSA+F, CSM, CSM+F, CS, and CS+F), ECp was 6–7% higher than in treatments with straw removed (SR and SR+F).

The net $CO_2$ exchange rate (NCER) can be described as a soil vitality index. We can conclude that the higher soil net $CO_2$ exchange rate depended more on the biological means used than on straw management. Treatments with ammonium nitrate revealed the lowest results compared to other treatments. A possible reason may be that the use of ammonium nitrate causes soil acidification [8]. Many microorganisms struggle to function properly under conditions of increased acidity. We assume that the reduced content of microorganisms caused a decrease in NCER. We found that NCER significantly depended on soil water content and crop nutrition conditions. The higher the water content found, the higher the NCER registered. The higher the ECp was determined, the higher the NCER was registered. This research confirmed that bioproducts applied favored soil vitality in general by exhibiting higher soil microbiological activity. As a result, a healthy and more viable *Cambisol* produced a higher grain yield in agricultural crops (Figure 18).

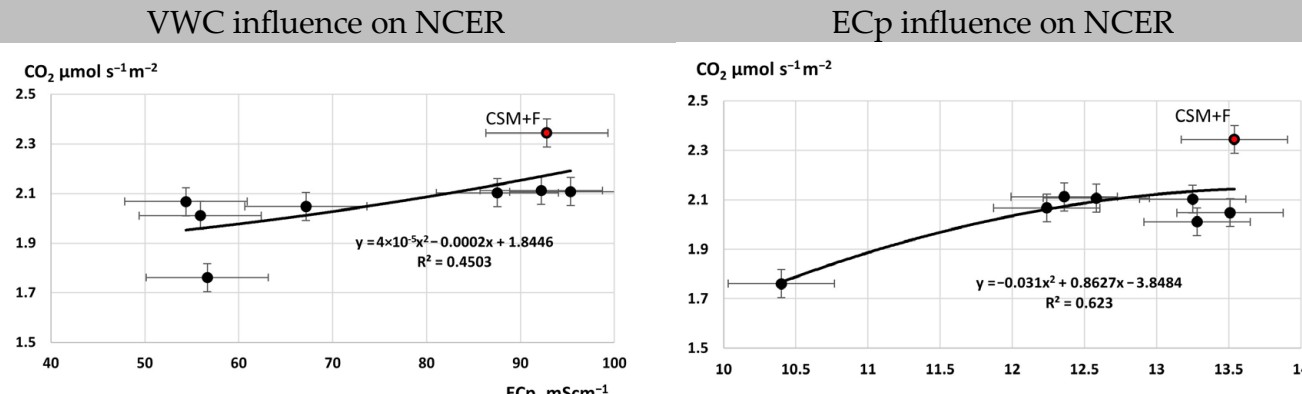

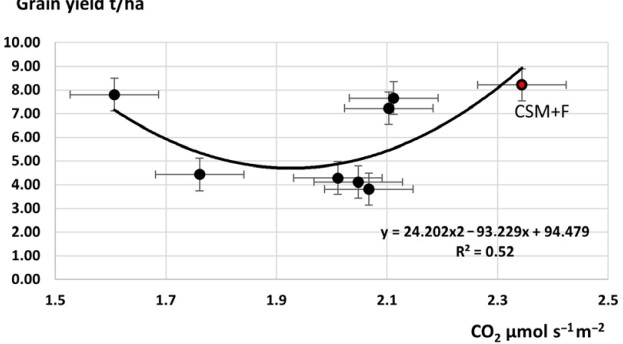

**Figure 18.** Soil sustainability and vitality dependence on soil management with microbiological products.

## 5. Conclusions

On Cambisol, with a shallow ploughless tillage background:

1. Chopped straw with amendments or without them improved soil pores electrical conductivity and total porosity, while straw acted as pore-clogging material within the overall arable layer, significantly decreasing the volume of mesopores. In treatments where straw was removed, the mean soil temperature was significantly higher than in treatments with straw return;

2. ECP was mostly dependent on mineral fertilizer application but independent of the straw management system. The effectiveness of biological products sprayed to promote straw mineralization was comparable to that of ammonium nitrate for the ECp index;

3. Straw management technology with biological products and complex fertilization was superior for microbial richness and biodiversity;

4. The use of complex fertilization had different effects on C sources in straw management technologies. Such fertilization in CSA+F treatment significantly increases the content of carbohydrates; in SCM+F treatment, the content of polymers, amino acids, and miscellaneous; in SR+F treatment, the content of carbohydrates, polymers, amines, and miscellaneous; and in CS+F treatment, the carboxylic acids and amino acids;

5. The winter wheat grain yield was mainly dependent on the application of mineral fertilizers. Bioproducts applied favored soil vitality by exhibiting higher soil microbiological activity. A more viable *Cambisol* produced a higher grain yield of wheat.

**Author Contributions:** Conceptualization, A.J. and D.F.; methodology, D.F., A.J. and V.F.; software, A.J., M.K., R.Ž. and D.F.; validation, A.J., D.F. and V.F.; formal analysis, A.J., M.K. and R.Ž.; investigation, A.J., D.F. and V.F.; writing—original draft preparation, A.J.; writing—review and editing, A.J., D.F. and V.F; visualization, A.J.; supervision, D.F. and V.F.; D.F., V.F. critically revised the work. All authors have read and agreed to the published version of the manuscript.

**Funding:** This work was partly supported by the research program "Productivity and sustainability of agricultural and forest soils" implemented by the Lithuanian Research Center for Agriculture and Forestry.

**Institutional Review Board Statement:** Not applicable.

**Informed Consent Statement:** Not applicable.

**Data Availability Statement:** The data that support the findings of this study are available from the first author, Arnoldas Jurys, upon reasonable request.

**Conflicts of Interest:** The authors declare no conflict of interest.

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
