# Peer review of "Aftereffect of Seven Years of Straw Handling on Soil Sustainability and Vitality"

_sustainability, doi:10.3390/su151712816_

Round 1

Reviewer 1 Report

The topic is of interest since it compares different agricultural treatments to measure the vitality of the crop soil.

I have the following observations:

1. Line 64- Methodology include the methods used to measure parameters such as NCER and others. Please describe it.

2. Lines 77, 88 and throughout the document- Clarify what is meant by fertilized, sometimes you say fertilized and in others NPI, please unify or explain it at the manuscripts so that it's clear. 

3. Line 77- Table 1, it could include the proportions of the straw, how much ammonium nitrate, and fertilizer was added, since it's not clear in the document.

4. Line 163- fig 6, the scale must be placed.

5. Line 236- fig 18, check figure numbering.

6. Line 262- fig 19, check figure numbering.

Author Response

Dear reviewer,
thank you for your comments.
We took into account your comments and adjusted the text accordingly.

Sincerely,

Arnoldas Jurys

Reviewer 2 Report

Review

Manuscript ID: Sustainability-2523704

“AFTEREFFECT OF SEVEN YEARS OF STRAW HANDLING ON SOIL SUSTAINABILITY AND VITALITY”

Properly treated straw can become a valuable source for soil improvement for the needs of crop nutrition. The goal of the study was to investigate the aftereffect of seven years use of mineral NPK fertilizers, bioproducts containing soil bacteria and microscopic fungi (Bacillus megaterium, Acinetobacter calcoaceticus and Trichoderma reesei) in combination with straw management on soil sustainability. The topic is of a practical nature, in the context of maintaining sustainability, which makes it fit well into the journal's profile.

However, the Introduction part is too short and not very comprehensive. I recommend to transfer some parts of the text from the results into the „Introduction.”

Materials and methods are described very carefully, and so are the results. But again, the Discussion part needs to be broaden, with more references (the same for the introduction). Also do not put in the discussion any figures – they should be in the results or in the materials and methods.

I recommend that you also refer more specific to the types of crops and the type of fertilization (in introduction and conclusions).

You should give more references to the scientific literature in this broad area, e.g.:

- Grygierzec B., Musiał K., Luty L. 2020. Sowing ratio, NS fertilisation and interactions of Lolium sp. and Festulolium grown in mixtures with Trifolium repens. Plant Soil Environ., 66 (8).

- Grygierzec B., Luty L., Musiał K. 2015. The efficiency of nitrogen and sulphur fertilization on yields and value of N:S ratio for Lolium x boucheanum. Plant Soil Environment. No. 3.

Also, the whole manuscript should be checked, in order to remove or correct some repetitions, typos, lack of space, unnecessary words, that occur in the text.

Author Response

Dear reviewer,
thank you for your comments.
We have adjusted the current version of the article based on your comments.

Sincerely,

Arnoldas Jurys

Reviewer 3 Report

The paper " AFTEREFFECT OF SEVEN YEARS OF STRAW HANDLING ON SOIL SUSTAINABILITY AND VITALITY" deals with an investigation of the uncertainties of the use of organic materials to improve soil health.

The article topic is engaging.

Some potential weaknesses and limitations of the article may include the limited scope of the study, as it only focuses on the impact of straw handling on soil sustainability and vitality in a specific type of soil (Cambisol) and under particular conditions (shallow plow-less tillage background). Additionally, the article may not consider other factors that could affect soil sustainability and vitality, such as climate, land use, and management practices. Therefore, some improvement is needed.

 Detailed comments.

 In the introduction: Case studies on the effect of STRAW HANDLING ON SOIL SUSTAINABILITY AND VITALITY should be included; similarly, the hypotheses and objectives at the end of this section.

Materials and Methods:

The experimental design of the research should be described in detail, showing the basic design, the structure of the treatments, the factors, the variables analyzed, the uncertainties, and the procedure for data analysis.

Line 71. The sentence “ley consisting of a mixture of 7 grasses” is not clear, please clarify the meaning of ley.

Results:

The results section should begin with the most important findings of the research according to the variables stated in the materials and methods.

A summary table of the mean squares of each variable in the Analysis of Variance (ANOVA) with its respective statistical significance should be placed. This will help to reduce the number of figures.

All figures should be structured according to the journal's guidelines for authors, error bars should be included, averages should not be included and, if possible, color should not be used, but the established conventions should be followed. Similarly, all figures and tables must be cited in the text.

All the results of the variables must be established or written in probabilistic terms, not only in terms of averages with higher and lower values.

In the paragraphs where the results of each variable are explained, it is not advisable to establish their conclusions; possible causal explanations for the results of the variables should be established. The whole manuscript is wordy. For instance, the words: Inner Mongolia and climate suitability appear more than seven times in the abstract and so on in the whole manuscript.

Discussion:

Lines 353 to 392 are redundant; the authors should begin by discussing why the most relevant findings were found in the stated variables.

Lines 382 – 383: Move to the introduction section.

The authors should make an in-depth discussion of each of the most relevant results, additionally, at the end of this section include the strengths, weaknesses, and limitations of applying this type of organic materials.

 Conclusions: Redo starting with the most representative findings.

Author Response

Dear reviewer,
thank you for your comments.
We have taken your comments into account while correcting the manuscript.

I'm sorry, but I didn't understand what you meant: "The whole manuscript is wordy. For instance, the words: Inner Mongolia and climate suitability appear more than seven times in the abstract and so on in the whole manuscript."

In response to your comment for: "Lines 353 to 392 are redundant; the authors should begin by discussing why the most relevant findings were found in the stated variables."

We don't agree. This part is important. It should be noted that microorganisms isolated from Lithuanian soil are used.

Sincerely,

Arnoldas Jurys

Reviewer 4 Report

Dear editor and authors,

This manuscript is really interesting. However, there are many points need to be revised so I would recommend this manuscript to be reconsidered after majorly visions.

For the first point, introduction part, this part needs to be more informative. It is still lacking of the problems of the statement and the concrete hypothesis and objectives. 

For the material and method part, this part is mostly lacking details of how you measure each parameter. I would recommend that you should look into details of each of the method that you used in your research.

For the result part, all bar chart and line graph need Error bars. Also, in the footnote or the captions of the figures and tables need more explanation Tables and figures need to be self- descriptive.

Author Response

  Dear reviewer, thank you for your comments.
We have taken your comments into account and added to the manuscript.

Sincerely,

Arnoldas Jurys

Round 2

Reviewer 1 Report

I agree with the changes incorporates.

Author Response

Thank you again for your comments and suggestions.

Sincerely,

Arnoldas Jurys

Reviewer 3 Report

The manuscript has been substantially improved. 

Author Response

(The authors gave the same response as above.)

Reviewer 4 Report

Dear Author, 

Thank you very much for your revised version. I satisfied with your revision.

Author Response

Dear Reviewer,

Thank you again for your comments and suggestions.

Sincerely,

Arnoldas Jurys